# Post-traumatic stress disorder and associated factors among road traffic accident survivors in Sub-Saharan Africa: A systematic review and meta-analysis

Tigabu Munye Aytenew[1]*, Getasew Legas[2], Solomon Demis Kebede[3], Amare Kassaw[4], Biruk Demissie[5], Adane Birhanu Nigat[1], Yirgalem Abere[6], Demewoz Kefale[4], Birhanu Mengist Munie[2]

1 Department of Nursing, College of Health Sciences, Debre Tabor University, Debre Tabor, Ethiopia, 2 Department of Psychiatry, College of Health Sciences, Debre Tabor University, Debre Tabor, Ethiopia, 3 Department of Maternity and Neonatal Nursing, College of Health Sciences, Debre Tabor University, Debre Tabor, Ethiopia, 4 Department of Pediatrics and Child Health Nursing, College of Health Sciences, Debre Tabor University, Debre Tabor, Ethiopia, 5 Department of Environmental Health, College of Health Sciences, Debre Tabor University, Debre Tabor, Ethiopia, 6 Department of Adult Health Nursing, College of Health Sciences, Debre Tabor University, Debre Tabor, Ethiopia

* tigabumunye21@gmail.com

## Abstract

### Introduction

Road traffic accidents have become a global public health issue, especially in low- and middle-income countries (LMICs). According to the World Health Organization (WHO) Global Road Safety Report 2018, there are over 1.35 million deaths related to road traffic accidents (RTAs) annually. Although several primary studies have been conducted to determine the prevalence and associated factors of post-traumatic stress disorder (PTSD) among RTA survivors in sub-Saharan Africa (SSA), these studies have reported inconsistent findings. Therefore, this study aimed to determine the pooled prevalence and associated factors of PTSD among RTA survivors in SSA.

### Methods

The studies were accessed through the Google Scholar, Scopus, PubMed, and Web of Science databases using search terms. Moreover, citation tracking was also performed. A random-effects DerSimonian-Laird model was used to compute the pooled prevalence of PTSD and determine associated factors among RTA survivors in SSA.

### Results

A total of 17 primary studies with a sample size of 9,056 RTA survivors were included in the final meta-analysis. The pooled prevalence of PTSD among RTA survivors in SSA was 23.36% (95% CI: 18.36, 28.36); $I^2 = 96.73\%$; $P < 0.001$). Female gender [AOR = 2.33, 95% CI: 1.80, 3.01], depression symptoms [AOR = 2.96, 95% CI: 2.17, 4.03], duration since the

**Data availability statement:** All relevant data are within the paper and its Supporting Information files.

**Funding:** The author(s) received no specific funding for this work.

**Competing interests:** The authors have declared that no competing interests exist.

**Abbreviations:** AOR: Adjusted odds ratio; CI: Confidence interval; JBI: Joanna Briggs Institute; LMICs: Low- and middle-income countries; PECO: Population, exposure, context and outcome; PRISMA: Preferred Reporting Items for Systematic Reviews and Meta-Analyses; PTSD: Post-traumatic stress disorder; RTA: Road traffic accident; SSA: Sub-Saharan Africa; WHO: World Health Organization.

accident (1-3 months) [AOR = 2.08, 95% CI: 1.23, 3.52], poor social support [AOR = 2.97, 95% CI: 1.09, 8.11], and substance use [AOR = 3.31, 95% CI: 1.68, 6.52] were significantly associated with PTSD.

## Conclusions

The pooled prevalence of PTSD was low in SSA compared to studies that have been conducted outside the region. Female gender, depression symptoms, duration since the accident (1-3 months), poor social support, and substance use were the pooled independent predictors of PTSD among RTA survivors in SSA. Those RTA survivors with these identified risk factors would be screened and managed early for PTSD using pharmacological treatment and brief psychological intervention. Future researchers shall conduct further studies using different methods, including qualitative studies to identify additional predictors of PTSD among RTA survivors in SSA.

## Introduction

Road traffic accident (RTA) is a crash originating from, terminating with, or involving a vehicle partially or fully on a public road, resulting in property damage, morbidity, and mortality [1]. RTAs have become a global public health issue, especially in low- and middle-income countries (LMICs). According to the World Health Organization (WHO) Global Road Safety Report 2018, there were over 1.35 million deaths related to RTAs annually [2]. RTA fatalities are predicted to be the second-leading cause of disability-adjusted life-years lost in developing countries, most of which were expected to occur in Africa by 2020 [3,4]. Currently, RTAs have become a global public health issue, especially in LMICs [5,6].

More than 90% of road traffic deaths occur in LMICs [6], and of this, sub-Saharan Africa (SSA) accounts for over 43% [7], and RTAs kill more people in SSA than malaria does [8]. RTAs can cause serious and long-lasting consequences for survivors, both in terms of physical and psychological outcomes including post-traumatic stress disorder (PTSD), depression symptoms, and anxiety [9–12]. In particular, PTSD is now a significant public health concern among RTA survivors [13,14].

Although most individuals exhibit PTSD symptoms within the first few weeks after trauma, more than 50% of patients improve without any intervention within three months [15]. Studies conducted among RTA survivors in the UK [16], USA [17], and Chinese Taiwan [18] reported that the prevalence of PTSD was 29.1%, 51%, and 82.2% respectively. Moreover, a systematic review and meta-analysis conducted among RTA survivors in Africa stated that the pooled prevalence of PTSD was 26% [14].

The severity of PTSD in people who witnessed or survived RTA depends on sex, age, place of injury, perceived life threat, responsibility for the injury, low income, lower educational level, comorbidity, history of mental illness, lack of social support, unemployment after the event, long-lasting physical problems following RTAs and property damage [19–24]. PTSD can cause significant behavioral changes that can lead to a loss of productivity, a loss of life, decreased quality of life, functional and socioeconomic deficits, and additional family disruptions [25–28].

There are substantial cultural variations and low socioeconomic levels in SSA compared to other regions of Africa, contributing to a significant health impact in the region. Moreover, although several primary studies have been conducted to determine the prevalence and associated factors of PTSD among RTA survivors in SSA, these studies have reported inconsistent findings. Therefore, this study aimed to determine the pooled prevalence and associated factors of PTSD among RTA survivors in SSA.

## Methods

### Reporting protocol

The Preferred Reporting Items for Systematic Reviews and Meta-Analyses (PRISMA) checklist [29] was used to report the findings of the study (S1 Table). The review protocol was registered with the Prospero database: (PROSPERO, 2024: CRD42024505402).

### Databases and search strategy

The adapted PECO format was used to retrieve the relevant studies. The adapted PECO consists of the population (P), exposure (E), context (C), and outcome (O) as detailed below.

a. **Population**: RTA survivors

b. **Exposure**: Associated factors, risk factors, determinants, and predictors, i.e., female gender, duration since accident (1-3 months), poor social support, comorbidity, near-misses, witnessing of death, substance use, depression symptoms, and family history of mental illness.

c. **Context (Setting)**: SSA, Ethiopia, Kenya, Uganda, Benin, Nigeria, and South Africa.

d. **Outcome:** PTSD among RTA survivors

Using the above adapted PECO, we developed the following review questions which focused on accessing all the relevant primary studies.

1. What is the prevalence of PTSD among RTA survivors in SSA?

2. What are the factors associated with PTSD among RTA survivors in SSA?

The studies were subsequently accessed through the Google Scholar, Scopus, PubMed, and Web of Science databases using the following search terms and phrases: ("Post-traumatic stress disorder [MeSH term] AND ("Predictors" [MeSH term] OR "Associated factors" [MeSH term] OR "Risk factors" [MeSH term] OR "Determinants" [MeSH term]) AND "Road traffic accident survivors" [MeSH term] AND "Sub-Saharan Africa"). The search string was developed using "AND" and "OR" Boolean operators. Moreover, citation tracking was also performed. The search was held from December 12 to 21/2023, and the searched studies were published between 2004 and 2023, and published in the English language.

### Eligibility criteria

All observational studies conducted among RTA survivors in SSA, reporting the prevalence of PTSD and/or the factor associated with PTSD, and written in English were included in the study. However, citations without abstracts, full texts, anonymous reports, editorials, systematic reviews, and meta-analyses were excluded from the study.

### Study selection

All the accessed studies were exported to the EndNote version 7 reference manager, and duplicate studies were removed. Initially, two independent reviewers (TMA and GL) screened the titles and abstracts, followed by the full-text reviews to determine the eligibility of each study. Discrepancies between the reviewers were resolved through dialog.

### Data extraction

Two independent reviewers (TMA and BMM) extracted the data using structured Microsoft Excel. When discrepancies or missing between the extracted data were detected, the phase was

repeated. When discrepancies between the data were continued, the third reviewer (GL) was involved. The name of the first author and year of publication, country, study design, sample size, measuring tool, response rate, and prevalence of the included studies were extracted.

### Outcome measures of interest

The primary outcome of interest was the prevalence of PTSD, and the second outcome of interest was factors affecting PTSD among RTA survivors in SSA.

### Operational definition of variables

The PTSD Checklist (PCL) civilian version, a self-report scale with 17 items having a five-point severity scale, was used to assess symptoms of PTSD. PTSD is defined if an individual reports at least one response of extremely severe symptoms in questions 1-5, at least one response of extremely severe symptoms in questions 6-12, and at least one response of extremely severe symptoms in questions 13–17 [30].

### Data analysis

STATA version 17 statistical software was used to analyze the statistical data. A random-effects DerSimonian-Laird model [31] was used to compute the pooled prevalence of PTSD and determine the impact of its associated factors. The publication bias was checked by examining the symmetry of the funnel plot, and Egger's test with a p-value of $< 0.05$ was used to determine significant publication bias [32]. The percentage of total variation across studies due to heterogeneity was assessed using $I^2$ statistics [33]. The values of $I^2$ 0, 25, 50, and 75% indicated no, low, moderate, and high heterogeneity respectively [33].

A p-value of the $I^2$ statistic $< 0.05$ was used to declare significant heterogeneity [34,35]. A sensitivity analysis was performed to identify the influence of a single study on the overall meta-analysis. A forest plot was generated to estimate the effect of independent factors on the outcome variable, and the 95% CI was calculated. The adjusted odds ratio (AOR) was the most frequently reported measure of association in the eligible primary studies.

## Results

### Search results

The search strategy identified a total of 896 studies from PubMed (461), Google Scholar (327), Scopus (25), and Web of Science database (83). After removing the irrelevant studies based on their titles and abstracts (n = 692) and duplicated studies (n = 51), a total of 153 studies were selected for full-text review.

Subsequently, full-text reviews were conducted, removing 124 studies due to different reasons (S1 Information). Then, 29 studies were assessed for full articles review and 12 studies were excluded (full texts were not written in English, conducted outside SSA, different target groups, and the outcomes were not well-defined). Finally, 17 studies were found to be relevant for determining the pooled prevalence of PTSD and identifying its associated factors. The PRISMA flow chart [36] was constructed to show the selection process from initially identified records to finally included primary studies (Fig 1).

### Characteristics of the included studies

Fourteen studies [8,37–49] and three [9,22,50] were conducted using cross-sectional and case-control respectively. Regarding geographical regions, seven studies [8,39,41,42,48–50] were conducted in Ethiopia, three [38,46,47] in South Africa, four [9,22,37,44] in Nigeria, one

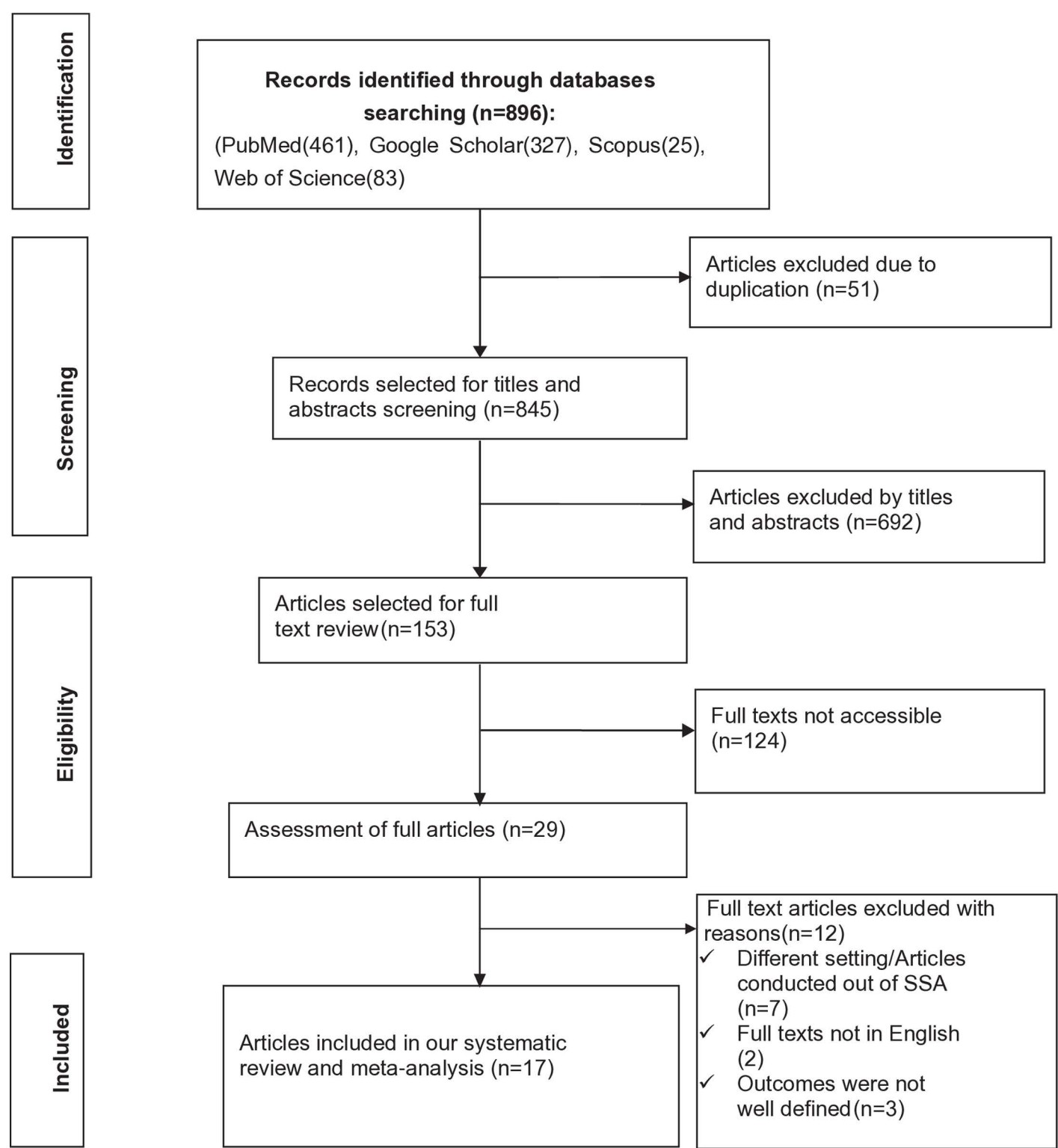

**Fig 1. PRISMA flow chart showing the studies selection process, 2024.**

[45] in Kenya, one [40] in Benin, and one [43] also in Uganda. The total sample size of the included studies was 9,056, where the smallest and largest sample sizes were 52 [46] and 4315 [38] among studies conducted in South Africa. The pooled prevalence of PTSD was obtained from the 16 included primary studies [8,9,22,37–49], whereas the data regarding the associated factors were obtained from the seven included primary studies [8,39,40,43,48–50]. The response rate of the included primary studies ranges from 85 to 100% (Table 1).

## Quality assessment of the included studies

Two independent reviewers (TMA and BMM) appraised the included studies' quality and scored the results' validity. The quality of each study was evaluated using the Joanna Briggs Institute (JBI) quality appraisal criteria [51]. Fourteen studies [8,37–49] and three [9,22,50] were appraised using the JBI checklist for cross-sectional and case-control respectively. Thus, among the fourteen cross-sectional studies, eleven scored seven of the eight questions, 87.5% (low risk), three scored six of the eight questions, 75% (low risk), and the remaining one study scored five of the eight questions, 62.5% (low risk). Likewise, among the three case-control studies, two scored eight of the ten questions, 80% (low risk), and the third scored seven of the ten questions, 70% (low risk). The cross-sectional studies scored between 5 and 7 out of 8 points, whereas the case-control studies scored between 7 and 8 out of 10 points (S2 Table). Studies were deemed low risk when they scored 50% or higher on the quality assessment indicators. Therefore, all the included primary studies [8,9,22,37–50] were of good quality.

## Risk of bias assessment

The adopted assessment tool [52] was used to assess the risk of bias. Accordingly, of the seventeen included primary studies, fourteen scored eight of the ten questions, two scored seven of the ten questions, and one scored six of the ten questions. Studies were classified as "low risk"

**Table 1. General characteristics of the included studies, 2024.**

| ID | Author [Year] | Study area | Study design | Sample size | Measuring tool | Prevalence (95% CI) | Quality |
|----|---------------|-----------|--------------|-------------|----------------|---------------------|---------|
| 1. | Ajibade BL [2015] [37] | Nigeria | CS | 400 | PCL | 51.80(46.90, 56.70) | Low risk |
| 2. | Alenko A [2019] [8] | Ethiopia | CS | 402 | TSQ | 12.60(9.36, 15.84) | Low risk |
| 3. | Asukuo JE [2017] [9] | Nigeria | Case-control | 92 | MINI | 41.30(31.24, 51.36) | Low risk |
| 4. | Atwoli L [2013] [38] | S/Africa | CS | 4315 | CIDI | 17.00(15.88, 18.12) | Low risk |
| 5. | Bedaso A [2020] [39] | Ethiopia | CS | 423 | PCL | 15.40(11.96, 18.84) | Low risk |
| 6. | Daddah D [2022] [40] | Benin | CS | 734 | PCL | 26.43(23.24, 29.62) | Low risk |
| 7. | Fekadu W [2019] [41] | Ethiopia | CS | 299 | PCL | 46.60(40.95, 52.25) | Low risk |
| 8. | Golja EA [2020] [42] | Ethiopia | CS | 193 | PCL | 17.10(11.79, 22.41) | Low risk |
| 9. | Isabirye RA [2022] [43] | Uganda | CS | 392 | MINI | 7.40(4.81, 9.99) | Low risk |
| 10. | Iteke O [2011] [22] | Nigeria | Case-control | 150 | MINI | 26.70(19.62, 33.78) | Low risk |
| 11. | Mosaku K [2014] [44] | Nigeria | CS | 121 | IES-R | 33.70(25.28, 42.12) | Low risk |
| 12. | Ongecha-Owuor FA, 2004 [45] | Kenya | CS | 264 | IES-R | 13.30(9.20, 17.40) | Low risk |
| 13. | Stein DJ [2016] [46] | S/Africa | CS | 52 | CIDI | 5.60(-0.65, 11.85) | Low risk |
| 14. | Suliman S [2014] [47] | S/Africa | CS | 131 | MINI | 19.60(12.80, 26.40) | Low risk |
| 15. | Tamirr TT [2022] [48] | Ethiopia | CS | 422 | CPSSI | 22.03(18.08, 25.98) | Low risk |
| 16. | Yimer GM [2023] [50] | Ethiopia | Case-control | 135 | PCL | Not applicable | Low risk |
| 17. | Yohannes K [2018] [49] | Ethiopia | CS | 531 | PCL | 22.80(19.23, 26.37) | Low risk |

Note: *CI, confidence interval; CS, cross-sectional*; CIDI, Composite International Diagnostic Interview; CPSSI, Child PTSD Symptom Scale Interview; IES-R, Revised Impact of Event Scale; MINI, Mini-International Neuro-psychiatric Interview; TSQ, Trauma Screening Questionnaire.

if eight and above of the ten questions received ″Yes″, as ″moderate risk″ if six to seven of the ten questions received ″Yes″ and ″high risk″ if five or lower of the ten questions received ″Yes″. Therefore, the three included primary studies [37,38,45] had some concerns of bias, and the fourteen primary studies [8,9,22,39–44,46–50] had a low risk of bias (high quality) (S3 Table).

## Meta-analysis

### Pooled prevalence of PTSD among RTA survivors

Finally, 17 eligible studies [8,9,22,37–50] were included in the final meta-analysis, and the pooled prevalence of PTSD among RTA survivors in SSA was 23.36% (95% CI: 18.36, 28.36); $I^2 = 96.73\%$; $P < 0.001$ (Fig 2).

### Publication bias

The asymmetry of the included primary studies on the funnel plot suggested the presence of publication bias (Fig 3a), and the p-value of Egger's test ($P = 0.0287$) also revealed this bias. Therefore, trim and fill analyses were performed to manage publication bias (Fig 3b).

### Investigation of heterogeneity

The percentage of $I^2$ statistics of the forest plot indicates marked heterogeneity among the included studies ($I^2 = 96.73\%$; $P < 0.001$). Hence, sensitivity and subgroup analyses were performed to investigate potential sources of heterogeneity.

### Sensitivity analysis

A sensitivity analysis was performed to determine the influence of a single study on the overall meta-analysis. The forest plot showed that the estimate of a single study was closer to the combined estimate, indicating the absence of a single study effect on the overall pooled estimate (Fig 4).

### Subgroup analysis

Subgroup analysis was performed using the study area and period. The subgroup analysis performed using the study area revealed that the highest pooled prevalence of PTSD was among studies conducted in Nigeria [38.50, 95% CI: 25.70, 51.30; $I^2 = 91.90\%$, $P < 0.001$], and the lowest pooled prevalence was among studies conducted in South Africa [14.27, 95%CI: 7.23, 21.31; $I^2 = 31.90\%$; $P < 0.001$]. Similarly, the subgroup analysis performed using the study period indicated that the higher pooled prevalence of PTSD was among studies conducted before 2020 [26.19, 95% CI: 18.98, 33.41; $I^2 = 97.19\%$; $P < 0.001$] followed by studies conducted in 2020 and after [17.64, 95% CI: 10.10, 25.17; $I^2 = 95.67\%$; $P < 0.001$] (Table 2). Based on the subgroup analyses, the heterogeneity of the study could be attributed to differences in the study area and period across the primary studies.

### Factors associated with PTSD among RTA survivors

Four studies [40,48–50] indicated that female gender was significantly associated with PTSD. The pooled AOR of PTSD for female gender was 2.33 (95% CI: 1.80, 3.01; $I^2 = 0.00\%$; $P < 0.85$) (Fig 5).

Five studies [8,39,43,49,50] showed a significant association between depression symptoms and PTSD. The pooled AOR of PTSD for RTA survivors with depression symptoms was 2.96 (95% CI: 2.17, 4.03; $I^2 = 0.00\%$; $P < 0.67$) (Fig 6).

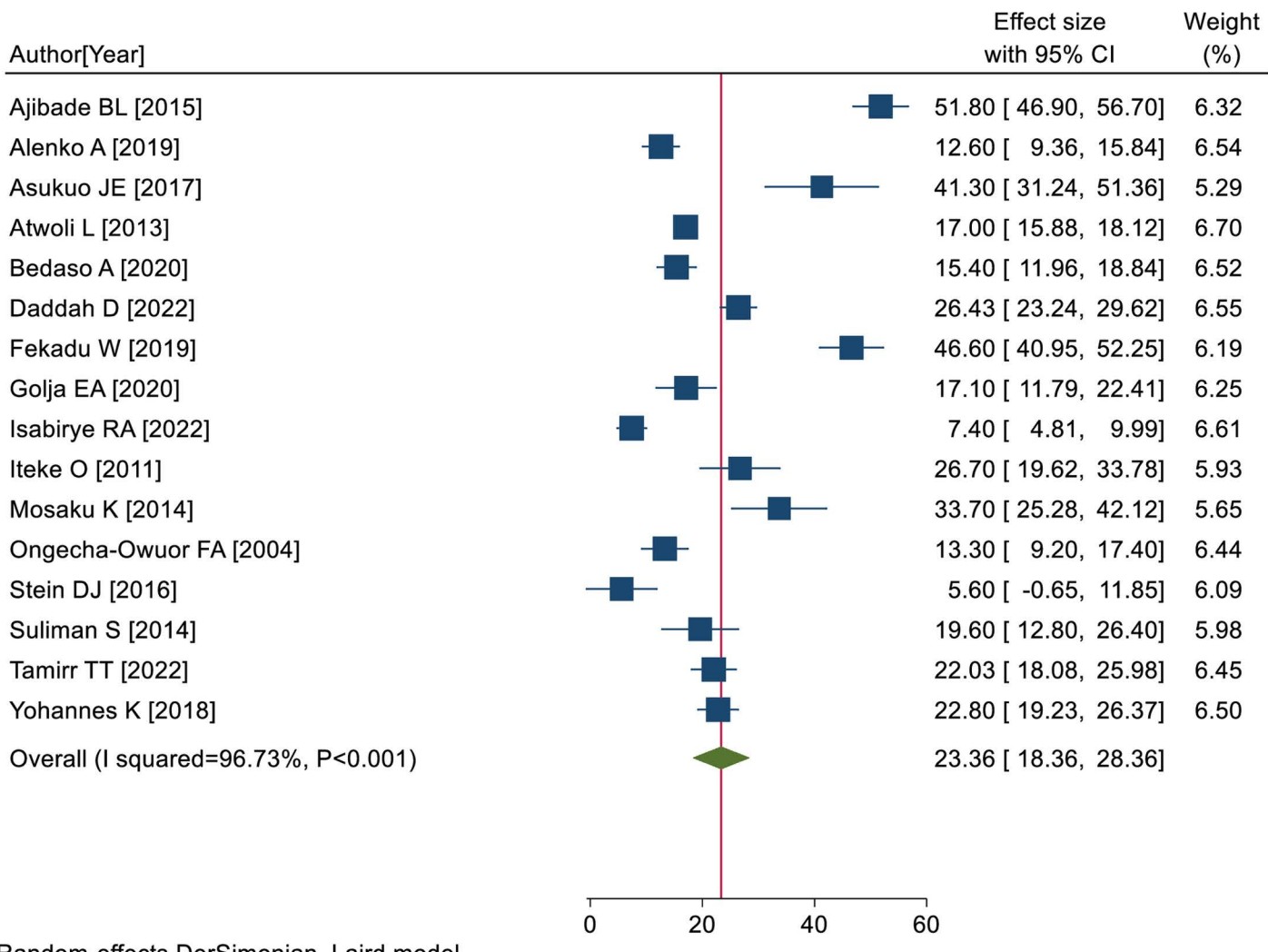

**Fig 2. Forest plot showing the pooled prevalence of PTSD with 95% CIs among RTA survivors in SSA, 2024.**

Two studies [39,49] reported a significant association between duration since the accident (1-3 months) and PTSD. The pooled AOR of PTSD for RTA survivors with duration since the accident (1-3 months) was 2.08 (95% CI: 1.23, 3.52; $I^2 = 30.93\%$; P < 0.23).

Three studies [48–50] showed that poor social support was significantly associated with PTSD. The pooled AOR of PTSD for RTA survivors with poor social support was 2.97 (95% CI: 1.09, 8.11; $I^2 = 83.99\%$; P < 0.001).

Two studies [8,43] revealed a significant association between substance use and PTSD. The pooled AOR of PTSD for RTA survivors with substance use was 3.31 (95% CI: 1.68, 6.52; $I^2 = 31.23\%$; P < 0.23).

## Discussion

The results of this study indicated that the pooled prevalence of PTSD was 23.36% (95% CI: 18.36, 28.36); $I^2 = 96.73\%$; P < 0.001). The results of this study were congruent with those studies conducted in Iran (19.2%) [53], Germany (18.4%) [54], and Africa (26.0%) [14]. The

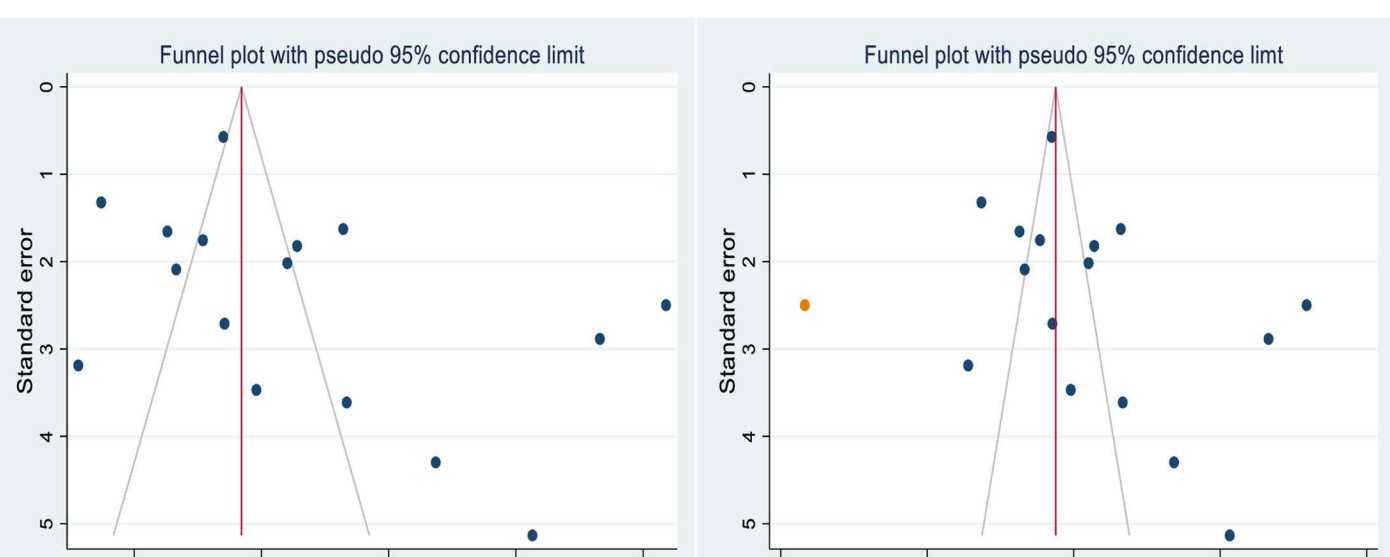

**Fig 3. Funnel plot showing the publication bias of PTSD among RTA survivors before adjustment (Fig 3a) and after adjustment with trim and fill analyses (Fig 3b) in SSA, 2024.**

results of this study, however, were less than those of studies carried out in the UK (29.1%) [16], California (41%) [55], USA (51%) [17], and Chinese Taiwan (82.2%) [18]. The time of PTSD assessment, the assessment tool used, and the cutoff point for diagnosis might be the most likely causes of this discrepancy among the included studies [39].

Furthermore, the results of this study revealed that female survivors had a 2.33 times higher risk of developing PTSD, whereas the estimation of the AORs could be inflated by the multicollinearity of repeated counting of cases in the included studies. The results of this study was supported by earlier studies conducted in Sweden [56], Croatia [57], and Australia [58]. It could be explained that women are less able to cope with stress than men [59–61].

Additionally, the results of this study showed that PTSD was 2.96 times more common in RTA survivors who also had depression symptoms. This could be because having pre-existing mental illnesses could increase the risk of developing PTSD [39,46]. Survivors with pre-existing mental illness have a lower quality of life, worse long-term health outcomes, and impaired physical functioning [62].

Likewise, this study's results showed that RTA survivors who had duration since the accident (1-3 months) were 2.08 times more likely to experience PTSD than those with duration since the accident of more than three months. The results of this study were aligned with that of a study conducted in California [63]. The likelihood of developing PTSD is primarily explained by the disease's short duration (within 3 months after the event) [39].

Additionally, the results of this study indicated that PTSD was 2.97 times more common in RTA survivors with poor social support than in those with good social support. The findings of this study were consistent with those studies conducted in Iran [53] and the UK [64]. This might be because a lack of social support after exposure to traumatic injury could lead to mental illness, and people with poor social support may not develop proper coping strategies after exposure to trauma [65].

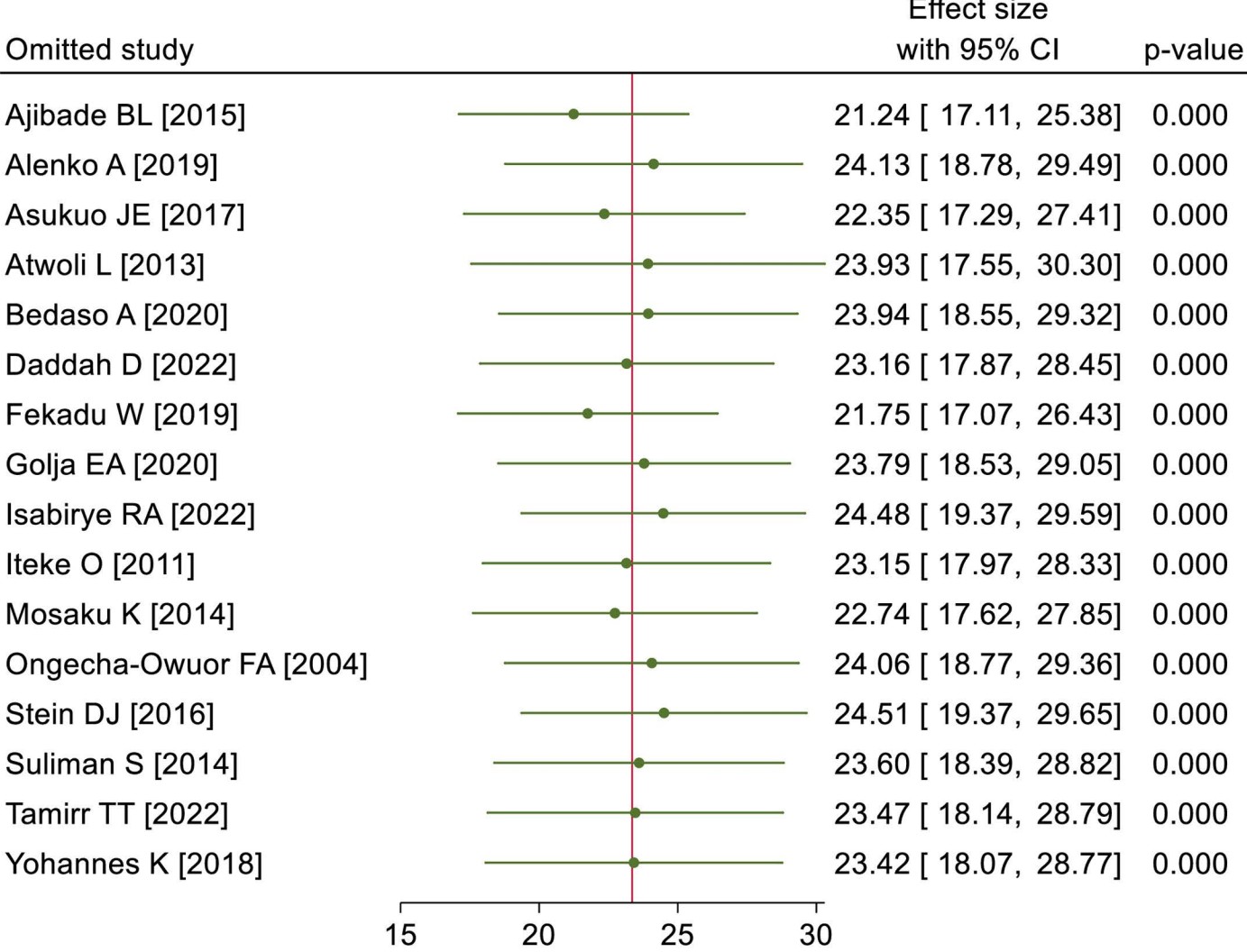

**Fig 4. Sensitivity analysis of PTSD among RTA survivors in SSA, 2024.**

**Table 2. Subgroup analyses of PTSD among RTA survivors in SSA, 2024.**

| Variables | Outcome | Subgroup | No. of studies | Model | Prevalence (95% CI) | I² | P-value |
|---|---|---|---|---|---|---|---|
| Study area | PTSD | Ethiopia | 6 | Random | 22.58(14.71, 30.45) | 95.70% | <0.0001 |
| | | Nigeria | 4 | Random | 38.50(25.70, 51.30) | 91.90% | <0.0001 |
| | | South Africa | 3 | Random | 14.27(7.23, 21.31) | 31.90% | <0.0001 |
| Study period | PTSD | <2020 | 11 | Random | 26.19(18.98, 33.41) | 97.19% | <0.0001 |
| | | ≥2020 | 5 | Random | 17.64(10.10, 25.17) | 95.67% | <0.0001 |

Note: CI, confidence interval; PTSD, post-traumatic stress disorder

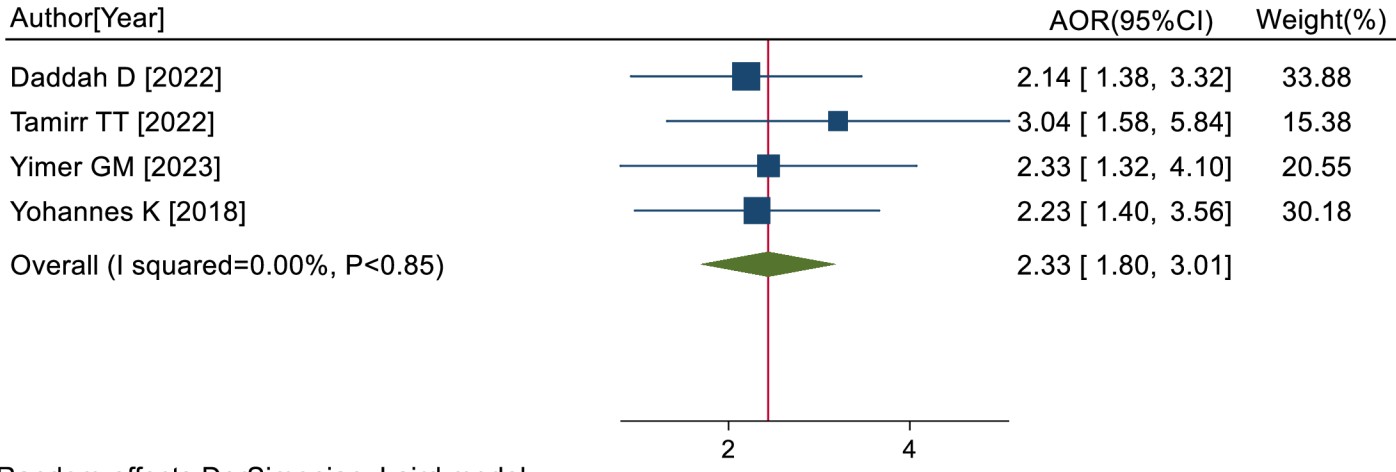

**Fig 5. Forest plot of the AORs with 95% CIs of studies on the association of female gender and PTSD among RTA survivors in SSA, 2024.**

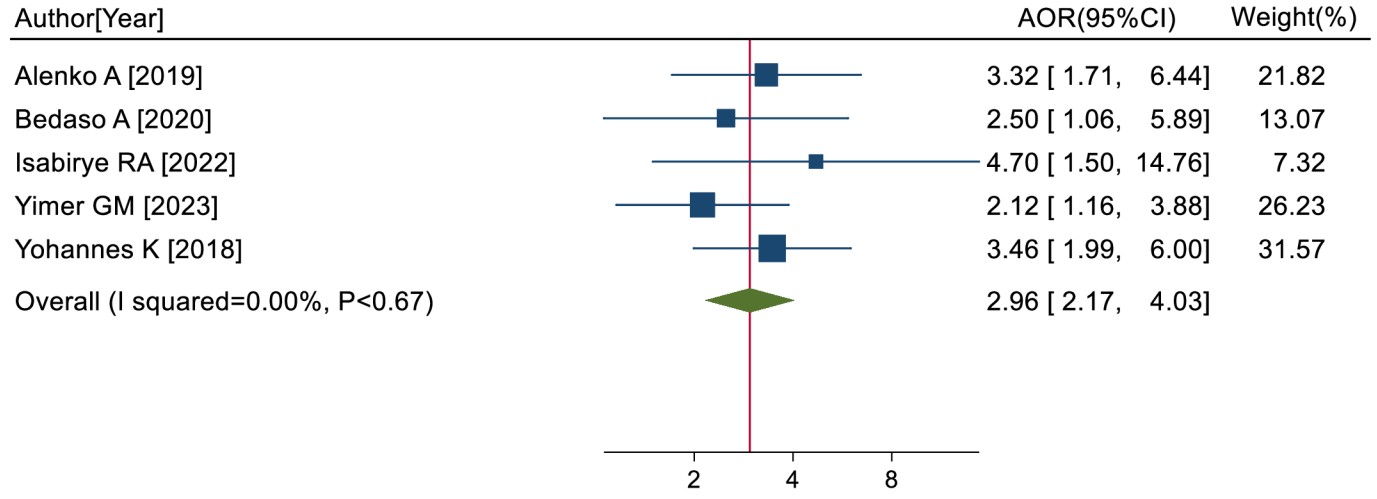

**Fig 6. Forest plot of the AORs with 95% CIs of studies on the association of depression symptoms and PTSD among RTA survivors in SSA, 2024.**

Moreover, the results of this study revealed substance using RTA survivors were 3.31 times higher risk of developing PTSD compared to their counterparts. RTA survivors with PTSD typically turn to substances as a coping mechanism for negative emotions like sadness and anxiety, intrusive memories and thoughts, hyperarousal, and trouble sleeping [66,67].

## Strengths and limitations of the study

As far as we know, this study was the first to compile the findings of multiple primary studies carried out in SSA, providing strong evidence on PTSD. Although all the included studies were of good quality, most of the studies were cross-sectional. The type of depression (pre-existing or comorbid type) was not clearly defined in some included studies, and we were

unable to demonstrate the potential source of heterogeneity even if we had performed sub-group analyses using the study area and period. Moreover, the majority of associated factors (duration since the accident, social support and substance use) lacked the minimum datasets needed for data synchronization in the meta-analysis, and the estimation of the AORs could be inflated by the multicollinearity of repeated counting of cases in the included studies.

## Conclusions

Compared to the studies carried out outside the region, the pooled prevalence of PTSD among RTA survivors was low in SSA. Female gender, depression symptoms, duration since the accident (1-3 months), poor social support, and substance use were the pooled independent predictors of PTSD among RTA survivors in SSA. Those RTA survivors with these identified risk factors would be screened and managed early for PTSD using pharmacological treatment and brief psychological intervention. Future researchers shall conduct further studies using a variety of methodologies, including qualitative studies to identify additional predictors of PTSD among RTA survivors in SSA.

## Supporting information

**S1 Information. Reasons for exclusion of studies.**
(DOCX)

**S1 Table. PRISMA checklist.**
(DOCX)

**S2 Table. Quality assessment of the included studies.**
(DOCX)

**S3 Table. Risk of bias assessment of the included studies.**
(DOCX)

## Acknowledgment

We would like to extend our deepest gratitude to Mr. Henok Andualem for his unreserved support throughout the study.

## Author contributions

**Conceptualization:** Tigabu Munye Aytenew.

**Data curation:** Tigabu Munye Aytenew.

**Formal analysis:** Tigabu Munye Aytenew, Amare Kassaw, Demewoz Kefale, Birhanu Mengist Munie.

**Methodology:** Tigabu Munye Aytenew, Getasew Legas, Solomon Demis Kebede, Adane Birhanu Nigat, Yirgalem Abere, Demewoz Kefale, Birhanu Mengist Munie.

**Resources:** Tigabu Munye Aytenew.

**Software:** Tigabu Munye Aytenew, Getasew Legas, Solomon Demis Kebede, Biruk Demissie, Adane Birhanu Nigat, Birhanu Mengist Munie.

**Validation:** Tigabu Munye Aytenew.

**Writing – original draft:** Tigabu Munye Aytenew.

**Writing – review & editing:** Tigabu Munye Aytenew, Getasew Legas, Solomon Demis Kebede, Amare Kassaw, Biruk Demissie, Yirgalem Abere, Birhanu Mengist Munie.

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
