## [Decision Letter · Decision Letter 0]

19 Nov 2024

PONE-D-24-41742Posttraumatic stress disorder and associated factors among road traffic accident survivors in Sub-Saharan Africa: A systematic review and meta-analysisPLOS ONE

Dear Dr. Aytenew,

Thank you for submitting your manuscript to PLOS ONE. After careful consideration, we feel that it has merit but does not fully meet PLOS ONE’s publication criteria as it currently stands. Therefore, we invite you to submit a revised version of the manuscript that addresses the points raised during the review process.

Two reviews have been obtained. Please find these below. There are major methodological issues that need to be addressed before this manuscript can be considered further.. 

We look forward to receiving your revised manuscript.

Kind regards,

Muhammad Haroon Stanikzai

Academic Editor

PLOS ONE

Journal Requirements:

https://bmcpublichealth.biomedcentral.com/counter/pdf/10.1186/s12889-024-19684-3.pdf?

https://karger.com/dem/article/doi/10.1159/000539449/909222/Predictors-of-Poststroke-Cognitive-Decline-among

In your revision ensure you cite all your sources (including your own works), and quote or rephrase any duplicated text outside the methods section. Further consideration is dependent on these concerns being addressed.

3. Please upload a new copy of Figures 1 and 3 to 7 as the detail is not clear. Please follow the link for more information:

https://blogs.plos.org/plos/2019/06/looking-good-tips-for-creating-your-plos-figures-graphics/
https://blogs.plos.org/plos/2019/06/looking-good-tips-for-creating-your-plos-figures-graphics/

4. As required by our policy on Data Availability, please ensure your manuscript or supplementary information includes the following: 

**Additional Editor Comments:**

-It would be preferable to change posttraumatic to post-traumatic through out the manuscript, including title.

-In the abstract add the abbreviations for low and middle-income countries (LMICs) and post-traumatic stress disorder (PTSD) (line 34). Please use the abbreviated form in later sections of the abstract.

- In the introduction, add the abbreviations for low and middle-income countries (LMICs) (line 71). Please use the abbreviated form in other parts of the introduction and other sections of the manuscript.

-Major point: The prevalence of PTSD may vary at different time aftermath of a trauma (e.g., 1 month, 3 months, 6 months, and 12 months). How do the authors address this issue? How late was PTSD assessed after RTA in the original studies? It would be preferable to address this issue in subgroup analysis.

-In Table 1, did all studies used PCL-civilian version for PTSD diagnosis?

-The discussion needs a bit more work. Recommend for each key observations. It is better to compare the findings of this systematic review with other systematic reviews. The authors have made comparisons mostly with cross-sectional studies.

-I also encourage you to copyedit your article by a native English speaker (particularly the discussion section).

Reviewers' comments:

Reviewer's Responses to Questions

**Comments to the Author**

1. Is the manuscript technically sound, and do the data support the conclusions?

Reviewer #1: Partly

Reviewer #2: Partly

2. Has the statistical analysis been performed appropriately and rigorously? 

Reviewer #1: Yes

Reviewer #2: No

3. Have the authors made all data underlying the findings in their manuscript fully available?

Reviewer #1: Yes

Reviewer #2: Yes

4. Is the manuscript presented in an intelligible fashion and written in standard English?

Reviewer #1: Yes

Reviewer #2: Yes

5. Review Comments to the Author

Reviewer #1: Thank you to the editorial team for inviting me to review this paper. I would also like to thank the authors for conducting research on such an important topic. I will provide a detailed review of the manuscript, offering suggestions, comments, and questions for improvement, line by line.

Abstract

1. Line 32: "Although several primary studies..." You mentioned "paucity of data" in line 36, which refers to scarcity. These statements appear contradictory. Could you consider rephrasing or clarifying your objective to resolve this?

Introduction

3. "RTAs have become a global public health issue...": The data cited is from seven years ago. If possible, please provide more recent statistics to update the reference to the 2017 WHO report.

4. Line 72: "More than 90% of road traffic deaths occur in low and middle-income countries..." Could you expand on the disparity between Sub-Saharan Africa and other regions?

5. Line 76: "In particular, PTSD is now a significant public health concern..." However, you cited references from 2006, 2015, and 2005, making it hard to support the "now" in your statement. Could you include more recent studies or reports to emphasize the current relevance of this concern, especially in the context of mental health trends in Sub-Saharan Africa?

6. While the introduction effectively outlines the issue, it could benefit from more focus on the factors driving the differences in PTSD rates globally.

Methods

7. Line 119: "The studies were accessed through..." Could you explain why these databases were chosen? Were others, such as African Journals Online, considered?

8. Did you pretest the search strategy in the databases before conducting the actual search? If so, could you mention what adjustments were made?

9. Line 132 (Eligibility criteria): Since the title is a systematic review and meta-analysis, could you clarify why qualitative studies were excluded?

10. Lines 141-146 (Data extraction) : Could you specify what tools were used to assess inter-rater reliability between the independent reviewers?

11. Lines 214-216 (Risk of bias assessment): You note that some studies raised concerns about the risk of bias but don’t elaborate on how this impacted the findings. A more detailed discussion here could strengthen this section.

Results

12. Search results: Why were 124 studies excluded after full-text review? Including a table with reasons for exclusion would improve transparency.

Discussion

13. Line 270: "The pooled prevalence of PTSD was 23.36%": You state that this is low compared to studies outside Africa but don’t explore why. Including a hypothesis or explanation would add depth to the discussion.

14.Lines 274-276: "The possible reasons for this difference..." Instead of listing possible reasons, could you elaborate on which explanation is most likely based on your data or literature?

15. Line 287: "Similarly, the findings of this study..." Did you consider analyzing data by age group or by rural versus urban settings? These factors could potentially influence PTSD rates.

Conclusion

16. Line 318: "Special attention should be given..." Could you be more specific about the types of interventions or attention required for these populations?

17. Lines 319-320: "Future researchers shall..." This is a strong recommendation, but providing concrete examples of triangulated study designs would strengthen it.

General comments

Some sections are dense with technical terms. Simplifying the language in key parts could make the manuscript more accessible to a broader audience.

Many of the references are outdated. Please consider updating them with more recent studies, including the latest WHO data.

Good luck!

Reviewer #2: The current article is a quantitative systematic review to summarise the evidence of the prevalence of post-road traffic accident PTSD and its potential risk factors. The research methodology basically follows the standard procedure of conducting systematic reviews. The results were interesting and timely. However, I have reservations about some details in the statistical analysis results, their interpretations, and article writing.

1. In Line 80 and Line 274, the authors cited and quoted a primary study from “Taiwan”. I would strongly recommend the author replace the term with “Chinese Taiwan”. I don’t mean to promote any political argument here, but please do consider how other Chinese scholars would cite this article if it were published. To avoid controversy in terms of storytelling and keep the reputation of journal, I would strongly recommend the authors follow the conduct of the UN.

2. In the “Databases and search strategy” section, the authors first used the term PECO in line 105, yet changed it to PICO in line 115. Although they are essentially the same, I would recommend you unify the term for better storytelling.

3. The “Primary outcome measure of interest” and “Operational definition of variables” sections only defined the outcome of the prevalence of PTSD. I wonder whether it is possible to add another section explaining how you define the outcomes of potential risk factors. Apparently, the authors counted the number of cases in later analysis to quantify each risk factor. It would be better to state this point clearly.

4. The main results of PTSD prevalence still show high heterogeneity after dividing by subgroups. Generally, such high heterogeneity indicates inconsistent results, as the authors suggested in the begining. It might be okay if the author failed to resolve high heterogeneity, but please address this as a limitation.

5. Following comment 3, the reason I strongly recommend the authors define how they quantify risk factors clearly is that the measurement of the risk factors might vary. For depression, for example, some studies you identified used PHQ-9, and some used Self-Reporting Questionnaire 20. Their criteria for diagnosing depression vary. Furthermore, please can you confirm whether the depression measured in each study was depression post-road traffic accident or pre-existing depression? And whether this is consistent across all studies? Depression is a common comorbid of PTSD, such that only pre-existing depression could be regarded as a risk factor (not to mention whether they had potential baseline PTSD). For example, in Fekadu 2018, the study population was adult survivors already. So, depression was measured among survivors of post-road traffic accidents, which is a comorbid of PTSD, not a risk factor. As a result, I have strong reservations about the authors’ results on depression as a risk factor.

a. Therefore, please also elaborate on the study summary table (table 1) regarding this information (case number in terms of sex, depression, means of measurement of risk factors, and so on)

b. I would strongly recommend the authors at least carefully discuss these points in the discussion section and state limitations.

6. Furthermore, the AOR estimated by the case numbers also suffered from the inflation caused by the multicollinearity of repeated counting of cases. That is, none of the studies distinguished whether the case had only one risk factor or several risk factors. For example, a female patient who had depression and substance use issues would be counted as a case for each factor. However, it is unclear which one was the key issue or whether there was interaction from other factors. This does not mean the results became completely unreliable, but please at least discuss this issue in the discussion section.

7. I have reservations about the latter 3 factors (duration since the accident, social support and substance use). Generally, it requires at least four datasets to perform data synchronisation in meta-analysis. None of these factors excessed 4. I would consider deleting them by saying there were not enough valid data or at least addressing the limitation.

8. Considering the above, the conclusion and abstract should also be amended accordingly.

6. PLOS authors have the option to publish the peer review history of their article (what does this mean? ). If published, this will include your full peer review and any attached files.

**Do you want your identity to be public for this peer review?** For information about this choice, including consent withdrawal, please see our Privacy Policy .

Reviewer #1: **Yes: ** Temesgen Anjulo Ageru

Reviewer #2: No

---

## [Author Response · Author response to Decision Letter 1]

25 Dec 2024

Dear Editors and reviewers:

We sincerely appreciate the valuable comments and suggestions you raised. The thorough review helped immensely in the shaping of the manuscript. The comments and suggestions have been closely followed and revisions have been made accordingly. The following are the questions that have been extracted from the Editor and Reviewers’ comments along with our summarized responses. Thank you very much for your constructive comments. We tried to inculcate your comments and questions as described below. The changes will be attached with

Title: Post-traumatic stress disorder and associated factors among road traffic accident survivors in Sub-Saharan Africa: A systematic review and meta-analysis

Authors:

TMA: tigabumunye21@gmail.com

GL: getasewlegas@gmail.com

SDK: solomondemis@gmail.com

AK: amarekassaw2009@gmail.com

BD: brookmelse2022@gmail.com

ABN: adanebirhanu23@gmail.com

YA: yirgalemabere12@gmail.com

DK: demewozk@yahoo.com

BMM: biremengist21@gmail.com

Editor Comment #01: Please ensure that your manuscript meets PLOS ONE's style requirements, including those for file naming. The PLOS ONE style templates can be found at

Authors’ response: Recognizing your comment, we have looked at the PLOS ONE style templates using the given link and we have ensured that our manuscript meets PLOS ONE's style requirements, including those for file naming. The requested corrections have been included throughout the revised version of the manuscript.

Editor Comment #02: We noticed you have some minor occurrence of overlapping text with the following previous publication(s), which needs to be addressed:

https://bmcpublichealth.biomedcentral.com/counter/pdf/10.1186/s12889-024-19684-3.pdf?

https://karger.com/dem/article/doi/10.1159/000539449/909222/Predictors-of-Poststroke-Cognitive-Decline-among

In your revision ensure you cite all your sources (including your own works), and quote or rephrase any duplicated text outside the methods section. Further consideration is dependent on these concerns being addressed.

In your revision ensure you cite all your sources (including your own works), and quote or rephrase any duplicated text outside the methods section.

Authors’ response: Accepting your constructive comment, we have paraphrased the overlapping texts throughout the manuscript.

Editor Comment #03: Please upload a new copy of Figures 1 and 3 to 7 as the detail is not clear. Please follow the link for more information: https://blogs.plos.org/plos/2019/06/looking-good-tips-for-creating-your-plos-figures-graphics/
https://blogs.plos.org/plos/2019/06/looking-good-tips-for-creating-your-plos-figures-graphics/

Authors’ response: Thank you! We have uploaded the figures with the required style.

Editor Comment #04: As required by our policy on Data Availability, please ensure your manuscript or supplementary information includes the following:

Confirmation that the study was eligible to be included in the review

If applicable for your analysis, a table showing the completed risks of bias and quality/certainty assessments for each study or outcome. Please ensure this is provided for each domain or parameter assessed. For example, if you used the Cochrane risk-of-bias tool for randomized trials, provide answers to each of the signaling questions for each study. If you used GRADE to assess certainty of evidence, provide judgments about each of the quality of evidence factor. This should be provided for each outcome.

Authors’ response: Recognizing your constructive feedback, we have uploaded the supplemental tables with their relevant information.

Editor Comment #05: It would be preferable to change posttraumatic to post-traumatic throughout the manuscript, including title.

Authors’ response: Thank you for your valuable feedback. We have changed posttraumatic to post-traumatic throughout the manuscript, including the title.

Editor Comment #06: In the abstract add the abbreviations for low and middle-income countries (LMICs) and post-traumatic stress disorder (PTSD) (line 34). Please use the abbreviated form in later sections of the abstract.

Authors’ response: Recognizing your feedback, we have revised the sessions.

Editor Comment #07: In the introduction, add the abbreviations for low and middle-income countries (LMICs) (line 71). Please use the abbreviated form in other parts of the introduction and other sections of the manuscript.

Authors’ response: Recognizing your feedback, we have revised the sessions.

Editor Comment #08: Major point: The prevalence of PTSD may vary at different time aftermath of a trauma (e.g., 1 month, 3 months, 6 months, and 12 months). How do the authors address this issue? How late was PTSD assessed after RTA in the original studies? It would be preferable to address this issue in subgroup analysis.

Authors’ response: Sure! The likelihood of developing PTSD is primarily explained within the short duration/the first months of the event/trauma. However, the majority of the included studies missed to report the prevalence of PTSD at 1st, 2nd, 3rd, 6th and 12th months of the event. These studies have reported only the lifelong PTSD following RTA.

Editor Comment #09: In Table 1, did all studies used PCL-civilian version for PTSD diagnosis?

Authors’ response: Thank you for your interesting question. The studies didn’t use only PCL-civilian version for PTSD diagnosis. I have revised the table by including all the diagnostic tools (CIDI, Composite International Diagnostic Interview; CPSSI, Child PTSD Symptom Scale Interview; IES-R, Revised Impact of Event Scale; MINI, Mini-International Neuro-psychiatric Interview; TSQ, Trauma Screening Questionnaire).

Editor Comment #10: The discussion needs a bit more work. Recommend for each key observations. It is better to compare the findings of this systematic review with other systematic reviews. The authors have made comparisons mostly with cross-sectional studies. I also encourage you to copyedit your article by a native English speaker (particularly the discussion section).

Authors’ response: Thank you for your constructive feedback! We have revised the manuscript, especially the discussion session by a language expert, and we have also compared this finding with previous findings of systematic review and meta-analysis.

Reviewer #1:

Reviewer #1 comment and suggestion #01: Abstract: Line 32: "Although several primary studies..." You mentioned "paucity of data" in line 36, which refers to scarcity. These statements appear contradictory. Could you consider rephrasing or clarifying your objective to resolve this?

Authors’ response: Thank you for your critical view of our manuscript! We initially perceived the "paucity of data" to mean multiple data, however, upon your valuable feedback, we have revised this session accordingly.

Reviewer #1 comment and suggestion #02: Introduction: "RTAs have become a global public health issue...” The data cited is from seven years ago. If possible, please provide more recent statistics to update the reference to the 2017 WHO report.

Authors’ response: Thank you for your critical view! We have used a recent report relatively.

Reviewer #1 comment and suggestion #03: Line 72: "More than 90% of road traffic deaths occur in low and middle-income countries..." Could you expand on the disparity between Sub-Saharan Africa and other regions?

Authors’ response: Thank you for your constructive feedback! We have mentioned the specific magnitude of PTSD among RTA survivors in SSA. .

Reviewer #1 comment and suggestion #04: Line 76: "In particular, PTSD is now a significant public health concern..." However, you cited references from 2006, 2015, and 2005, making it hard to support the "now" in your statement. Could you include more recent studies or reports to emphasize the current relevance of this concern, especially in the context of mental health trends in Sub-Saharan Africa?

Authors’ response: Thank you very much for your critical view! We have used recent references on their behalf.

Reviewer #1 comment and suggestion #05: While the introduction effectively outlines the issue, it could benefit from more focus on the factors driving the differences in PTSD rates globally.

Authors’ response: Thank you! In addition to narrating the magnitude and burden of PTSD among RTA survivors, we have also narrated the potential risk factors of PTSD among RTA survivors.

Reviewer #1 comment and suggestion #06: Line 119: "The studies were accessed through..." Could you explain why these databases were chosen? Were others, such as African Journals Online, considered?

Authors’ response: Thank you! We have chosen these search engines and databases because we could easily access all the relevant studies. We have also searched African Journals Online and Google to retrieve unpublished articles from their repositories.

Reviewer #1 comment and suggestion #07: Did you pretest the search strategy in the databases before conducting the actual search? If so, could you mention what adjustments were made?

Authors’ response: Yes! Initially we have used ʺPost-traumatic stress disorder AND ʺSub-Saharan Africaʺ MeSH terms to retrieve the relevant studies as a pretest. Finally, we have used ʺPost-traumatic stress disorder [MeSH term] AND (ʺPredictorsʺ [MeSH term] OR ʺAssociated factorsʺ [MeSH term] OR ʺRisk factorsʺ [MeSH term] OR ʺDeterminantsʺ [MeSH term]) AND ʺRoad traffic accident survivorsʺ [MeSH term] AND ʺSub-Saharan Africaʺ) to retrieve the relevant studies.

Reviewer #1 comment and suggestion #08: Line 132 (Eligibility criteria): Since the title is a systematic review and meta-analysis, could you clarify why qualitative studies were excluded?

Authors’ response: Thank you for your critical view! Even though we have mentioned it as exclusion criteria, unfortunately, we didn’t exclude a qualitative study from this study, and we have revised this session too.

Reviewer #1 comment and suggestion #09: Lines 141-146 (Data extraction): Could you specify what tools were used to assess inter-rater reliability between the independent reviewers?

Authors’ response: Thank you! Two independent reviewers (TMA and BMM) extracted the data using structured Microsoft Excel independently, and cross-checking was held each other. When discrepancies or missing between the extracted data were detected, the phase was repeated. When discrepancies between the data were continued again, the third reviewer (NE) was involved.

Reviewer #1 comment and suggestion #10: Lines 214-216 (Risk of bias assessment): You note that some studies raised concerns about the risk of bias but don’t elaborate on how this impacted the findings. A more detailed discussion here could strengthen this section.

Authors’ response: Thank you for your constructive feedback! We have performed the risk of bias assessment in detail, and attached as supplemental table.

Reviewer #1 comment and suggestion #11: Results: Search results: Why were 124 studies excluded after full-text review? Including a table with reasons for exclusion would improve transparency.

Authors’ response: Thank you! We have included a supplemental table stating the reasons why these studies have been excluded.

Reviewer #1 comment and suggestion #12: Discussion: Line 270: "The pooled prevalence of PTSD was 23.36%": You state that this is low compared to studies outside Africa but don’t explore why. Including a hypothesis or explanation would add depth to the discussion.

Authors’ response: Recognizing your valuable comment, we have elaborated it in the discussion session why this discrepancy occurred.

Reviewer #1 comment and suggestion #13: Lines 274-276: "The possible reasons for this difference..." Instead of listing possible reasons, could you elaborate on which explanation is most likely based on your data or literature?

Authors’ response: Thank you! We have elaborated the most likely reasons.

Reviewer #1 comment and suggestion #14: Line 287: "Similarly, the findings of this study..." Did you consider analyzing data by age group or by rural versus urban settings? These factors could potentially influence PTSD rates.

Authors’ response: Thank you! We have faced a difficulty to compare the data based on age group and rural versus urban because the majority of these studies didn’t have aggregate report PTSD in terms of these variables.

Reviewer #1 comment and suggestion #15: Line 318: "Special attention should be given..." Could you be more specific about the types of interventions or attention required for these populations?

Authors’ response: Recognizing your constructive comment, we have specified the specific intervention strategies.

Reviewer #1 comment and suggestion #16: Lines 319-320: "Future researchers shall..." This is a strong recommendation, but providing concrete examples of triangulated study designs would strengthen it.

Authors’ response: Thank you for your valuable feedback! We have also revised this session specifically.

Reviewer #1 comment and suggestion #17: General comments: Some sections are dense with technical terms. Simplifying the language in key parts could make the manuscript more accessible to a broader audience.

Many of the references are outdated. Please consider updating them with more recent studies, including the latest WHO data.

Authors’ response: Thank you! We have revised the whole manuscript by consulting language expert, and used updated references relatively.

Reviewer #2:

Reviewer #2 comment and suggestion #01: In Line 80 and Line 274, the authors cited and quoted a primary study from “Taiwan”. I would strongly recommend the author replace the term with “Chinese Taiwan”. I don’t mean to promote any political argument here, but please do consider how other Chinese scholars would cite this article if it were published. To avoid controversy in terms of storytelling and keep the reputation of journal, I would strongly recommend the authors follow the conduct of the UN.

Authors’ response: Recognizing your feedback, we have revised these sessions.

Reviewer #2 comment and suggestion #02: In the “Databases and search strategy” section, the authors first used the term PECO in line 105, yet changed it to PICO in line 115. Although they are essentially the same, I would recommend you unify the term for better storytelling.

Authors’ response: Thank you for your critical view! It was typo, and we have revised it.

Reviewer #2 comment and suggestion #03: The “Primary outcome measure of interest” and “Operational definition of variables” sections only defined the outcome of the prevalence of PTSD. I wonder whether it is possible to add another section explaining how you define the outcomes of potential risk factors. Apparently, the authors counted the number of cases in later analysis to quantif

---

## [Decision Letter · Decision Letter 1]

12 Jan 2025

PONE-D-24-41742R1Post-traumatic stress disorder and associated factors among road traffic accident survivors in Sub-Saharan Africa: A systematic review and meta-analysisPLOS ONE

Dear Dr. Aytenew,

Thank you for submitting your manuscript to PLOS ONE. After careful consideration, we feel that it has merit but does not fully meet PLOS ONE’s publication criteria as it currently stands. Therefore, we invite you to submit a revised version of the manuscript that addresses the points raised during the review process.

Thank you for addressing the initial comments provided by the reviewers. Based on the revised manuscript, the reviewers have requested that some minor comments still need to be addressed.

We look forward to receiving your revised manuscript.

Kind regards,

Muhammad Haroon Stanikzai

Academic Editor

PLOS ONE

Journal Requirements:

Additional Editor Comments :

- Line 45: Please replace depression with depression symptoms. Please consider this through out the manuscript.

- Line 60: Please spell out RTA at first use in the main body of the manuscript.

- Line 62: Please spell out LMICs and WHO at first use in the main body of the manuscript.

- Line 70: Please spell out PTSD at first use in the main body of the manuscript.

Reviewers' comments:

Reviewer's Responses to Questions

**Comments to the Author**

1. If the authors have adequately addressed your comments raised in a previous round of review and you feel that this manuscript is now acceptable for publication, you may indicate that here to bypass the “Comments to the Author” section, enter your conflict of interest statement in the “Confidential to Editor” section, and submit your "Accept" recommendation.

Reviewer #1: All comments have been addressed

Reviewer #2: (No Response)

2. Is the manuscript technically sound, and do the data support the conclusions?

Reviewer #1: Yes

Reviewer #2: No

3. Has the statistical analysis been performed appropriately and rigorously? 

Reviewer #1: Yes

Reviewer #2: Yes

4. Have the authors made all data underlying the findings in their manuscript fully available?

Reviewer #1: Yes

Reviewer #2: Yes

5. Is the manuscript presented in an intelligible fashion and written in standard English?

Reviewer #1: Yes

Reviewer #2: Yes

6. Review Comments to the Author

Reviewer #1: All my previous comments, suggestions, and questions were answered. I don't have further questions, comments and suggestions.

Reviewer #2: 1. In line 297, the following was revised: “Furthermore, even though the AOR estimated by the case numbers could be suffered from the inflation caused by the multicollinearity of repeated counting of cases in the included studies, the results of this study revealed that female survivors had a 2.33 times higher risk of developing PTSD, according to the study’s findings…” The author indeed attempted to suggest the point that “case numbers could be suffered from the inflation…” but with “even though”. This, in English, seems to tell the reader that the next result is not a problem. I am not sure whether this is what the author intends to express because, in fact, there is a problem with the results. That is, the 2.33 times odds are inflated and so inaccurate (not incorrect but inaccurate). The author needs to be very careful about making such a statement. All the odds that the author estimated here are not absolute values but only relative values. Instead, the author should consider stating that, for example, the current data suggest that female survivors had a 2.33 times higher risk of developing PTSD, whereas the estimation of the AORs would be inflated by…and thus, the results need to be interpreted with caution.

2. The author needs to address the overestimation of AORs in the limitation. It is scientifically sound to say these factors are associated, but these estimated AORs are not. This is important. (for example, someone could quote this paper like: the author said women are twice as likely to have PTSD. But that is not accurate. It is because women are more likely to get depression. Depression holds up most of the variance of PTSD. So maybe being female is related to PTSD, but that 2.33 times odds is not the precise number to suggest. And so do all other estimated AORs.)

7. PLOS authors have the option to publish the peer review history of their article (what does this mean? ). If published, this will include your full peer review and any attached files.

**Do you want your identity to be public for this peer review?** For information about this choice, including consent withdrawal, please see our Privacy Policy .

Reviewer #1: **Yes: ** Temesgen Anjulo Ageru

Reviewer #2: No

---

## [Author Response · Author response to Decision Letter 2]

13 Jan 2025

Dear Editors and reviewers:

We sincerely appreciate the valuable comments and suggestions you raised. The thorough review helped immensely in the shaping of the manuscript. The comments and suggestions have been closely followed and revisions have been made accordingly. The following are the questions that have been extracted from the Editor and Reviewers’ comments along with our summarized responses. Thank you very much for your constructive comments. We tried to inculcate your comments and questions as described below. The changes will be attached with

Title: Post-traumatic stress disorder and associated factors among road traffic accident survivors in Sub-Saharan Africa: A systematic review and meta-analysis

Authors:

TMA: tigabumunye21@gmail.com

GL: getasewlegas@gmail.com

SDK: solomondemis@gmail.com

AK: amarekassaw2009@gmail.com

BD: brookmelse2022@gmail.com

ABN: adanebirhanu23@gmail.com

YA: yirgalemabere12@gmail.com

DK: demewozk@yahoo.com

BMM: biremengist21@gmail.com

Editor Comment #01: Please review your reference list to ensure that it is complete and correct. If you have cited papers that have been retracted, please include the rationale for doing so in the manuscript text, or remove these references and replace them with relevant current references. Any changes to the reference list should be mentioned in the rebuttal letter that accompanies your revised manuscript. If you need to cite a retracted article, indicate the article’s retracted status in the References list and also include a citation and full reference for the retraction notice.

Authors’ response: Recognizing your comment, we have explored and ensured that the reference lists are complete, and we have not cited a retracted articled in the study.

Editor Comment #02: Line 45: Please replace depression with depression symptoms. Please consider this throughout the manuscript.

Authors’ response: Accepting your constructive comment, we have replaced depression with depression symptoms throughout the manuscript.

Editor Comment #03: Line 60: Please spell out RTA at first use in the main body of the manuscript

Authors’ response: Thank you for your constructive feedback! We have spelled out RTA at first use in the main body of the manuscript.

Editor Comment #04: Line 62: Please spell out LMICs and WHO at first use in the main body of the manuscript.

Authors’ response: Recognizing your constructive feedback, we have spelled out LMICs and WHO at first use in the main body of the manuscript.

Editor Comment #05: Line 70: Please spell out PTSD at first use in the main body of the manuscript.

Authors’ response: Thank you for your valuable feedback! We have spelled out PTSD at first use in the main body of the manuscript too.

Reviewer #2:

Reviewer #2 comment and suggestion #01: In line 297, the following was revised: “Furthermore, even though the AOR estimated by the case numbers could be suffered from the inflation caused by the multicollinearity of repeated counting of cases in the included studies, the results of this study revealed that female survivors had a 2.33 times higher risk of developing PTSD, according to the study’s findings…” The author indeed attempted to suggest the point that “case numbers could be suffered from the inflation…” but with “even though”. This, in English, seems to tell the reader that the next result is not a problem. I am not sure whether this is what the author intends to express because, in fact, there is a problem with the results. That is, the 2.33 times odds are inflated and so inaccurate (not incorrect but inaccurate). The author needs to be very careful about making such a statement. All the odds that the author estimated here are not absolute values but only relative values. Instead, the author should consider stating that, for example, the current data suggest that female survivors had a 2.33 times higher risk of developing PTSD, whereas the estimation of the AORs would be inflated by…and thus, the results need to be interpreted with caution.

Authors’ response: Recognizing your constructive feedback, we have revised this session accordingly.

Reviewer #2 comment and suggestion #02: The author needs to address the overestimation of AORs in the limitation. It is scientifically sound to say these factors are associated, but these estimated AORs are not. This is important. (for example, someone could quote this paper like: the author said women are twice as likely to have PTSD. But that is not accurate. It is because women are more likely to get depression. Depression holds up most of the variance of PTSD. So maybe being female is related to PTSD, but that 2.33 times odds is not the precise number to suggest. And so do all other estimated AORs.

Authors’ response: Thank you for your valuable feedback again! We have also addressed it in the limitation section of the manuscript.

---

## [Decision Letter · Decision Letter 2]

21 Jan 2025

Post-traumatic stress disorder and associated factors among road traffic accident survivors in Sub-Saharan Africa: A systematic review and meta-analysis

PONE-D-24-41742R2

Dear Dr. Aytenew,

We’re pleased to inform you that your manuscript has been judged scientifically suitable for publication and will be formally accepted for publication once it meets all outstanding technical requirements.

Kind regards,

Muhammad Haroon Stanikzai

Academic Editor

PLOS ONE

Additional Editor Comments (optional):

The revised manuscript demonstrates significant improvements and addresses all the previously highlighted concerns. The authors have carefully considered the reviewers' feedback and have made appropriate amendments, enhancing the clarity, accuracy, and overall quality of the work.

Reviewers' comments:

Reviewer's Responses to Questions

**Comments to the Author**

1. If the authors have adequately addressed your comments raised in a previous round of review and you feel that this manuscript is now acceptable for publication, you may indicate that here to bypass the “Comments to the Author” section, enter your conflict of interest statement in the “Confidential to Editor” section, and submit your "Accept" recommendation.

Reviewer #2: All comments have been addressed

2. Is the manuscript technically sound, and do the data support the conclusions?

Reviewer #2: Partly

3. Has the statistical analysis been performed appropriately and rigorously? 

Reviewer #2: Yes

4. Have the authors made all data underlying the findings in their manuscript fully available?

Reviewer #2: Yes

5. Is the manuscript presented in an intelligible fashion and written in standard English?

Reviewer #2: Yes

6. Review Comments to the Author

Reviewer #2: (No Response)

7. PLOS authors have the option to publish the peer review history of their article (what does this mean? ). If published, this will include your full peer review and any attached files.

**Do you want your identity to be public for this peer review?** For information about this choice, including consent withdrawal, please see our Privacy Policy .

Reviewer #2: No

---

## [Editor Report · Acceptance letter]

PONE-D-24-41742R2

PLOS ONE

Dear Dr. Aytenew,

I'm pleased to inform you that your manuscript has been deemed suitable for publication in PLOS ONE. Congratulations! Your manuscript is now being handed over to our production team.

Kind regards,

on behalf of

Dr. Muhammad Haroon Stanikzai

Academic Editor

PLOS ONE